# GroupMamba: Parameter-Efficient and Accurate Group Visual State Space Model

## Abstract

Recent advancements in state-space models (SSMs) have showcased effective performance in modeling long-range dependencies with subquadratic complexity. However, pure SSM-based models still face challenges related to stability and achieving optimal performance on computer vision tasks. Our paper addresses the challenges of scaling SSM-based models for computer vision, particularly the instability and inefficiency of large model sizes. To address this, we introduce a Modulated Group Mamba layer which divides the input channels into four groups and applies our proposed SSM-based efficient Visual Single Selective Scanning (VSSS) block independently to each group, with each VSSS block scanning in one of the four spatial directions. The Modulated Group Mamba layer also wraps the four VSSS blocks into a channel modulation operator to improve cross-channel communication. Furthermore, we introduce a distillation-based training objective to stabilize the training of large models, leading to consistent performance gains. Our comprehensive experiments demonstrate the merits of the proposed contributions, leading to superior performance over existing methods for image classification on ImageNet-1K, object detection, instance segmentation on MS-COCO, and semantic segmentation on ADE20K. Our tiny variant with 23M parameters achieves state-of-the-art performance with a classification top-1 accuracy of 83.3% on ImageNet-1K, while being 26% efficient in terms of parameters, compared to the best existing Mamba design of same model size. Our code and models will be publicly released.

## 1 Introduction

Various context modeling methods have emerged in the domains of language and vision understanding. These include Convolution (He et al., 2016; Yang et al., 2022), Attention (Vaswani et al., 2017), and, more recently, State Space Models Gu et al. (2022); Gu & Dao (2023). Transformers with their multi-headed self-attention mechanism (Vaswani et al., 2017) have been central to both language models such as GPT-3 (Brown et al., 2020) and vision models such as Vision Transformers (Dosovitskiy et al., 2021; Liu et al., 2021). However, challenges arose due to the quadratic computational complexity of attention mechanisms particularly for longer sequences, leading to the recent emergence of State Space models such as S4 (Gu et al., 2022).

While being effective in handling extended input sequences due to their linear complexity in terms of sequence lengths, S4 (Gu et al., 2022) encountered limitations in global context processing in information-dense data, especially in domains like computer vision due to the data-independent nature of the model. Alternatively, approaches such as global convolutions-based state space models (Fu et al., 2023b) and Liquid S4 (Hasani et al., 2022) have been proposed to mitigate the aforementioned limitations. The recent Mamba (Gu & Dao, 2023) introduces the S6 architecture which aims to enhance the ability of state-space models to handle long-range dependencies efficiently. The selective-scan algorithm introduced by Mamba uses input-dependent state-space parameters, which allow for better in-context learning while still being computationally efficient compared to self-attention.

However, Mamba, specifically the S6 algorithm, is known to be unstable for e.g., image classification, especially when scaled to large sizes (Patro & Agneeswaran, 2024). Additionally, the Mamba model variant used in image classification, generally called the VSS (Visual State Space) block, can be more efficient in terms of parameters and compute requirements based on the number of channels. The VSS block includes extensive input and output projections along with depth-wise convolutions,

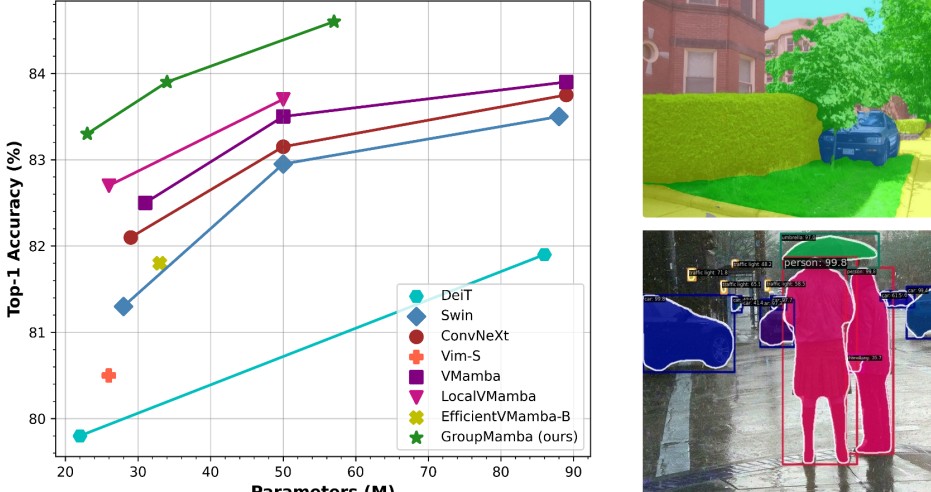

Figure 1: **Left:** Comparison in terms of Parameters vs. Top-1 Accuracy on ImageNet-1k (Deng et al., 2009). Our GroupMamba-B achieves superior top-1 classification accuracy while reducing parameters by 36% compared to VMamba (Liu et al., 2024b). **Right:** Qualitative results of GroupMamba-T on semantic segmentation (top right), and object detection and instance segmentation (bottom right). More qualitative examples are presented in Figure 3 and the supplemental material.

whose parameters and compute complexities are directly proportional to the number of channels in the input. To address this issue, we propose a *Modulated Group Mamba* layer that mitigates the aforementioned issues in a computation and parameter-efficient manner. The main contributions of our paper are:

1. We introduce a *Modulated Group Mamba* layer, inspired by Group Convolutions, which enhances computational efficiency and interaction in state-space models by using a multi-direction scanning method for comprehensive spatial coverage and effective modeling of local and global information.

2. We introduce a *Channel Affinity Modulation (CAM)* operator, which enhances communication across channels to improve feature aggregation, addressing the limited interaction inherent in the grouping operation.

3. To address the instability issue in the SSM-based architecture, we introduce a distillation-based training objective designed to stabilize models with a large number of parameters, leading to better performance and a smooth loss convergence trend.

4. We build a series of parameter-efficient generic classification models called "GroupMamba", based on the proposed *Modulated Group Mamba* layer. Our *tiny* variant achieves 83.3% top-1 accuracy on ImageNet-1k (Deng et al., 2009) with $23M$ parameters and $4.5G$ FLOPs. Additionally, our *base* variant achieves top-1 accuracy of 84.5% with $57M$ parameters and $14G$ FLOPs, outperforming all recent SSM methods (see Figure 1).

## 2 RELATED WORK

Convolutional Neural Networks (ConvNets) have been the popular choice for computer vision tasks since the introduction of AlexNet (Krizhevsky et al., 2012). The field has rapidly evolved with several landmark ConvNet architectures (Simonyan & Zisserman, 2015; Szegedy et al., 2015; He et al., 2016; Howard et al., 2017; Tan & Le, 2019). Alongside these architectural advances, significant efforts have been made to refine individual convolution layers, including depthwise convolution (Xie et al., 2017), group convolution (Cohen & Welling, 2016), and deformable convolution (Dai et al., 2017). Recently, ConvNeXt variants (Liu et al., 2022b; Woo et al., 2023) have taken concrete steps towards

modernizing traditional 2D ConvNets by incorporating macro designs with advanced settings and training recipes to achieve on-par performance with the state-of-the-art models.

In recent years, the pioneering Vision Transformer (ViT) (Dosovitskiy et al., 2021) has significantly impacted the computer vision field, including tasks such as image classification (Touvron et al., 2021; Liu et al., 2021; 2022a; Fan et al., 2021), object detection (Carion et al., 2020; Zhu et al., 2021; Meng et al., 2021; Zhang et al., 2022), and segmentation (Cheng et al., 2022; Shaker et al., 2024; Kirillov et al., 2023). ViT (Dosovitskiy et al., 2021) introduces a monolithic design that approaches an image as a series of flattened 2D patches without image-specific inductive bias. The remarkable performance of ViT for computer vision tasks, along with its scalability, has inspired numerous subsequent endeavors to design better architectures. The early ViT-based models usually require large-scale datasets (e.g., JFT-300M (Sun et al., 2017)) for pretraining. Later, DeiT (Touvron et al., 2021) proposes advanced training techniques in addition to integrating a distillation token into the architecture, enabling effective training on smaller datasets (e.g., ImageNet-1K (Deng et al., 2009)). Since then, subsequent studies have designed hierarchical and hybrid architectures by combining CNN and ViT modules to improve performance on different vision tasks (Srinivas et al., 2021; Maaz et al., 2022; d'Ascoli et al., 2021; Shaker et al., 2023; Fan et al., 2021). Another line of work is to mitigate the quadratic complexity inherent in self-attention, a primary bottleneck of ViTs. This effort has led to significant improvements and more efficient and approximated variants (Wang et al., 2020; Shaker et al., 2023; Pan et al., 2022; Mehta & Rastegari, 2023; Kitaev et al., 2020; Chu et al., 2021; Tu et al., 2022), offering reduced complexity while maintaining effectiveness.

Recently, State Space Models (SSMs) have emerged as an alternative to ViTs (Vaswani et al., 2017), capturing the intricate dynamics and inter-dependencies within language sequences (Gu et al., 2022). One notable method in this area is the structured state-space sequence model (S4) (Gu et al., 2022), designed to tackle long-range dependencies while maintaining linear complexity. Following this direction, several models have been proposed, including S5 (Smith et al., 2023), H3 (Fu et al., 2023a), and GSS (Mehta et al., 2022). More recently, Mamba (Gu & Dao, 2023) introduces an input-dependent SSM layer and leverages a parallel selective scan mechanism (S6).

In the visual domain, various works have applied SSMs to different tasks. In particular for image classification, VMamba (Liu et al., 2024b) uses Mamba with bidirectional scans across both spatial dimensions in a hierarchical Swin-Transformer (Liu et al., 2021) style design to build a global receptive field efficiently. A concurrent work, Vision Mamba (Vim) (Zhu et al., 2024), instead proposed a monolithic design with a single bidirectional scan for the entire image, outperforming traditional vision transformers like DeiT. LocalVMamba (Huang et al., 2024) addresses the challenge of capturing detailed local information by introducing a scanning methodology within distinct windows (inspired from Swin-Transformer (Liu et al., 2021)), coupled with dynamic scanning directions across network layers. EfficientVMamba (Pei et al., 2024) integrates atrous-based selective scanning and dual-pathway modules for efficient global and local feature extraction, achieving competitive results with reduced computational complexity. These models have been applied for image classification, as well as image segmentation (Liu et al., 2024a; Ma et al., 2024; Ruan & Xiang, 2024; Gong et al., 2024), video understanding (Yang et al., 2024; Li et al., 2024; Chen et al., 2024), and various other tasks (Guo et al., 2024b; He et al., 2024; Wang et al., 2024; Guo et al., 2024a; Liang et al., 2024). Their wide applicability shows the effectiveness of SSMs (Gu et al., 2022; Smith et al., 2023; Fu et al., 2023a; Mehta et al., 2022), and in particular Mamba (Gu & Dao, 2023), in the visual domain. In this paper, we propose a *Modulated Group Mamba* layer that mitigates the drawbacks of the default vision Mamba block, such as lack of stability (Patro & Agneeswaran, 2024) and the increased number of parameters with respect to the number of channels.

## 3 METHOD

**Motivation:** Our method is motivated based on the observations with respect to the limitations of existing Visual State-Space models.

- *Lack of Stability for Larger Models*: We observe from Patro & Agneeswaran (2024) that Mamba (Gu & Dao, 2023) based image classification models with an MLP channel mixer are unstable when scaled to a large number of parameters. This instability can be seen in SiMBA-L (MLP) (Patro & Agneeswaran, 2024), which leads to sub-optimal classification results of 49% accuracy. We mitigate this issue by introducing a *Modulated Group Mamba*

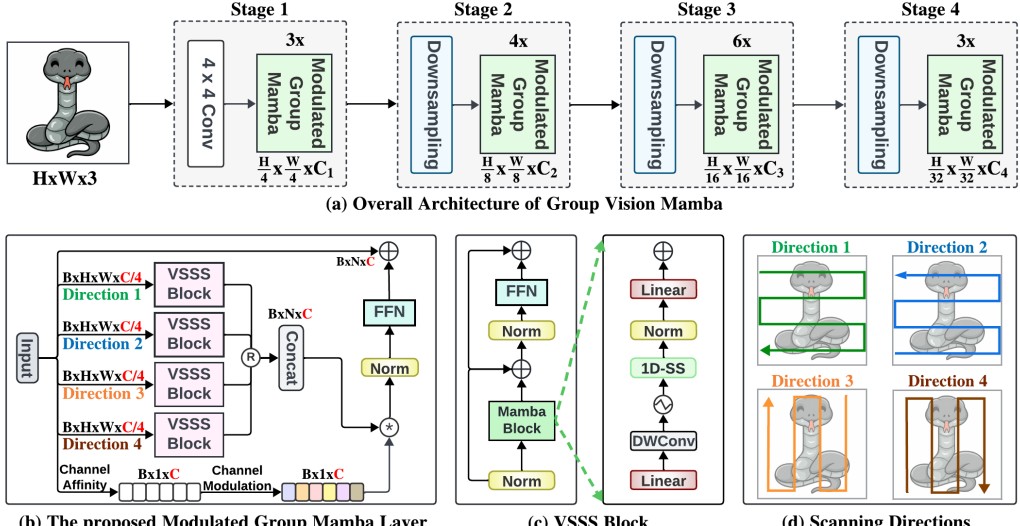

Figure 2: Overview of the proposed method. **Top Row:** The overall architecture of our framework with a consistent hierarchical design comprising four stages. **Bottom Row:** We present **(b)** The design of the modulated group mamba layer. The input channels are divided into four groups with a single scanning direction for each VSSS block. This significantly reduces the computational complexity compared to the standard mamba layer, with similar performance. Channel Affinity Modulation mechanism is introduced to address the limited interactions within the VSSS blocks. **(c)** The design of VSSS block. It consists of Mamba block with 1D Selective Scanning block followed by FFN. **(d)** The four scanning directions used for the four VSSS blocks are illustrated.

design alongside a distillation objective (as presented in Section 3.4) that stabilizes the Mamba SSM training without modifying the channel mixer.

- *Efficient Improved Interaction*: Given the computational impact of Mamba-based design on the number of channels, the proposed *Modulated Group Mamba* layer is computationally inexpensive and parameter efficient than the default Mamba and able to model both local and global information from the input tokens through multi-direction scanning. An additional *Channel Affinity Modulation* operator is proposed in this work to compensate for the limited channel interaction due to the grouped operation.

## 3.1 PRELIMINARIES

**State-Space Models:** State-space models (SSMs) like S4 (Gu et al., 2022) and Mamba (Gu & Dao, 2023) are structured sequence architectures inspired by a combination of recurrent neural networks (RNNs) and convolutional neural networks (CNNs), with linear or near-linear scaling in sequence length. Derived from continuous systems, SSMs define and 1D *function-to-function map* for an input $x(t) \in \mathbb{R}^L \to y(t) \in \mathbb{R}^L$ via a hidden state $h(t) \in \mathbb{R}^N$. More formally, SSMs are described by the continuous time Ordinary Differential Equation (ODE) in Equation 1.

$$h'(t) = \mathbf{A}h(t) + \mathbf{B}x(t),$$
$$y(t) = \mathbf{C}h(t),$$
(1)

where $h(t)$ is the current hidden state, $h'(t)$ is the updated hidden state, $x(t)$ is the current input, $y(t)$ is the output, $\mathbf{A} \in \mathbb{R}^{N \times N}$ is SSM's evolution matrix, and $\mathbf{B} \in \mathbb{R}^{N \times 1}, \mathbf{C} \in \mathbb{R}^{N \times 1}$ are the input and output projection matrices, respectively.

**Discrete State-Space Models:** To allow these models to be used in sequence modeling tasks in deep learning, they need to be discretized, converting the SSM from a continuous time *function-to-function map* into a discrete-time *sequence-to-sequence map*. S4 (Gu et al., 2022) and Mamba (Gu & Dao, 2023) are among the discrete adaptations of the continuous system, incorporating a timescale parameter $\boldsymbol{\Delta}$ to convert the continuous parameters $\mathbf{A}, \mathbf{B}$ into their discrete equivalents $\overline{\mathbf{A}}, \overline{\mathbf{B}}$. This

discretization is typically done through the Zero-Order Hold (ZOH) method given in Equation 2.

$$\begin{aligned}
\overline{\mathbf{A}} &= \exp(\mathbf{\Delta A}), \\
\overline{\mathbf{B}} &= (\mathbf{\Delta A})^{-1}(\exp(\mathbf{\Delta A}) - \mathbf{I}) \cdot \mathbf{\Delta B} \\
h_t &= \overline{\mathbf{A}} h_{t-1} + \overline{\mathbf{B}} x_t, \\
y_t &= \mathbf{C} h_t.
\end{aligned} \quad (2)$$

While both S4 (Gu et al., 2022) and Mamba (Gu & Dao, 2023) utilize a similar discretization step as stated above in Equation 2, Mamba differentiates itself from S4 by conditioning the parameters $\mathbf{\Delta} \in \mathbb{R}^{B \times L \times D}$, $\mathbf{B} \in \mathbb{R}^{B \times L \times N}$ and $\mathbf{C} \in \mathbb{R}^{B \times L \times N}$, on the input $x \in \mathbb{R}^{B \times L \times D}$, through the S6 Selective Scan Mechanism, where $B$ is the batch size, $L$ is the sequence length, and $D$ is the feature dimension.

## 3.2 Overall Architecture

As shown in Figure 2 (a), our model uses a hierarchical architecture, similar to Swin Transformer (Liu et al., 2021), with four stages to efficiently process images at varying resolutions. Assuming an input image, $\mathbf{I} \in \mathbb{R}^{H \times W \times 3}$, we first apply a Patch Embedding layer to divide the image into non-overlapping patches of size $4 \times 4$ and embed each patch into a $C_1$-dimensional feature vector. The patch embedding layer is implemented using two $3 \times 3$ convolutions with a stride of 2. This produces features maps of size $\frac{H}{4} \times \frac{W}{4} \times C_1$ at the first stage. These feature maps are passed to a stack of our Modulated Grouped Mamba blocks (as detailed in Section 3.3). In each subsequent stage, a down-sampling layer merges patches in a $2 \times 2$ region, followed by another stack of our Modulated Grouped Mamba blocks. Hence, feature size at stages two, three and four are $\frac{H}{8} \times \frac{W}{8} \times C_2$, $\frac{H}{16} \times \frac{W}{16} \times C_3$, and $\frac{H}{32} \times \frac{W}{32} \times C_4$, respectively.

## 3.3 Modulated Group Mamba Layer

We present the overall operations of the proposed *Modulated Group Mamba* layer (Figure 2 (b)) for an input sequence $\mathbf{X}_{\text{in}}$, with dimensions $(B, H, W, C)$, where $B$ is the batch size, $C$ is the number of input channels and $H/W$ are the width and height of the feature map, in Equation 3.

$$\begin{aligned}
\mathbf{X}_{\text{GM}} &= \text{GroupedMamba}(\mathbf{X}_{\text{in}}, \Theta) \\
\mathbf{X}_{\text{CAM}} &= \text{CAM}(\mathbf{X}_{\text{GM}}, \text{Affinity}(\mathbf{X}_{\text{in}})) \\
\mathbf{X}_{\text{out}} &= \mathbf{X}_{\text{in}} + \text{FFN}(\text{LN}(\mathbf{X}_{\text{CAM}}))
\end{aligned} \quad (3)$$

Here, $\mathbf{X}_{\text{GM}}$ is the output of Equation 6, $\mathbf{X}_{\text{CAM}}$ is the output of Equation 9, LN is the Layer Normalization (Ba et al., 2016) operation, FFN is the Feed-Forward Network as described by Equation 5, and $\mathbf{X}_{\text{out}}$ is the final output of the Modulated Group Mamba block. The individual operations, namely the GroupedMamba operator, the VSSS block used inside the GroupedMamba operator, and the CAM operator, are presented in Section 3.3.1, Section 3.3.2 and Section 3.3.3, respectively.

### 3.3.1 Visual Single Selective Scan (VSSS) Block

The VSSS block (Figure 2 (c)) is a token and channel mixer based on the Mamba operator. Mathematically, for an input token sequence $\mathbf{Z}_{\text{in}}$, the VSSS block performs the operations as described in Equation 4.

$$\begin{aligned}
\mathbf{Z}'_{\text{out}} &= \mathbf{Z}_{\text{in}} + \text{Mamba}(\text{LN}(\mathbf{Z}_{\text{in}})) \\
\mathbf{Z}_{\text{out}} &= \mathbf{Z}'_{\text{out}} + \text{FFN}(\text{LN}(\mathbf{Z}'_{\text{out}}))
\end{aligned} \quad (4)$$

Where $\mathbf{Z}_{\text{out}}$ is the output sequence, Mamba is the discretized version of the Mamba SSM operator as described in Equation 2.

$$\text{FFN}(\text{LN}(\mathbf{Z}'_{\text{out}})) = \text{GELU}(\text{LN}(\mathbf{Z}'_{\text{out}})\mathbf{W}_1 + \mathbf{b}_1)\mathbf{W}_2 + \mathbf{b}_2 \quad (5)$$

Where GELU (Hendrycks & Gimpel, 2016) is the activation function and $\mathbf{W}_1$, $\mathbf{W}_2$, $\mathbf{b}_1$, and $\mathbf{b}_2$ are weights and biases for the linear projections.

### 3.3.2 GROUPED MAMBA OPERATOR

Considering the motivation presented earlier in Section 3, we aim to design a variant of the Mamba (Gu & Dao, 2023) that is both computationally efficient and can effectively model the spatial dependencies of the input sequence. Given that Mamba is computationally inefficient on large number of channels $C$ in the input sequence, we propose a grouped variant of the operator, inspired by Grouped Convolutions. The Grouped Mamba operation is a variant of the VSSS block presented in Section 3.3.1, where the input channels are divided into groups, and the VSSS operator is applied separately to each group. Specifically, we divide the input channels into four groups, each of size $\frac{C}{4}$, and an independent VSSS block is applied to each group. To better model spatial dependencies in the input, each of the four groups scans in one of four directions across the token sequence: left-to-right, right-to-left, top-to-bottom, and bottom-to-top, as outlined in Figure 2 (d).

Let $G = 4$ be the number of groups representing four scanning directions: left-to-right, right-to-left, top-to-bottom, and bottom-to-top. We form four sequences from the input sequence $\mathbf{X}_{\text{in}}$, namely $\mathbf{X}_{\text{LR}}$, $\mathbf{X}_{\text{RL}}$, $\mathbf{X}_{\text{TB}}$, and $\mathbf{X}_{\text{BT}}$, each of shape $(B, H, W, \frac{C}{4})$, representing one of the four directions specified earlier. These are then flattened to form a single token sequence of shape $(B, N, \frac{C}{4})$, where $N = W \times H$ is the number of tokens in the sequence. The parameters for each of the four groups can be specified by $\theta_{\text{LR}}$, $\theta_{\text{RL}}$, $\theta_{\text{TB}}$, and $\theta_{\text{BT}}$, respectively, for each of the four groups, representing the parameters for the VSSS blocks.

Given the above definitions, the overall relation for the Grouped Mamba operator can be written as shown in Equation 6.

$$\begin{aligned}
\mathbf{X}_{\text{GM}} = \text{GroupedMamba}(\mathbf{X}_{\text{in}}, \Theta) = \text{Concat}(&\text{VSSS}(\mathbf{X}_{\text{LR}}, \Theta_{\text{LR}}), \\
&\text{VSSS}(\mathbf{X}_{\text{RL}}, \Theta_{\text{RL}}), \\
&\text{VSSS}(\mathbf{X}_{\text{TB}}, \Theta_{\text{TB}}), \\
&\text{VSSS}(\mathbf{X}_{\text{BT}}, \Theta_{\text{BT}}))
\end{aligned} \tag{6}$$

Where:

- $\mathbf{X}_{\text{LR}}$, $\mathbf{X}_{\text{RL}}$, $\mathbf{X}_{\text{TB}}$, and $\mathbf{X}_{\text{BT}}$ represent the input tensors scanned in the respective directions.

- $\Theta_{\text{LR}}$, $\Theta_{\text{RL}}$, $\Theta_{\text{TB}}$, and $\Theta_{\text{BT}}$ represents the parameters of the VSSS block for each direction.

- The output of each Mamba operator is reshaped again to $(B, H, W, \frac{C}{4})$, and concatenated back to form the token sequence $\mathbf{X}_{\text{GM}}$, again of the size $(B, H, W, C)$.

### 3.3.3 CHANNEL AFFINITY MODULATION (CAM)

On its own, the Grouped Mamba operator may have a disadvantage in the form of limited information exchange across channels, given the fact that each operator in the group only operates over $\frac{C}{4}$ channels. To encourage the exchange of information across channels, we propose a Channel Affinity Modulation operator, which recalibrates channel-wise feature responses to enhance the representation power of the network. In this block, we first average pool the input to calculate the channel statistics as shown in Equation 7.

$$\text{ChannelStat}(\mathbf{X}_{\text{in}}) = \text{AvgPool}(\mathbf{X}_{\text{in}}) \tag{7}$$

where $\mathbf{X}_{\text{in}}$ is the input tensor, and AvgPool represents the global average pooling operation. Next comes the affinity calculation operation as shown in Equation 8.

$$\text{Affinity}(\mathbf{X}_{\text{in}}) = \sigma\left(W_2 \delta\left(W_1 \text{ChannelStat}(\mathbf{X}_{\text{in}})\right)\right) \tag{8}$$

where $\delta$ and $\sigma$ represent non-linearity functions, and $W_1$ and $W_2$ are learnable weights. The role of $\sigma$ is to assign an importance weight to each channel to compute the affinity. The result of the affinity calculation is used to recalibrate the output of the Grouped Mamba operator, as shown in Equation 9.

$$\mathbf{X}_{\text{CAM}} = \text{CAM}(\mathbf{X}_{\text{GM}}, \text{Affinity}(\mathbf{X}_{\text{in}})) = \mathbf{X}_{\text{GM}} \cdot \text{Affinity}(\mathbf{X}_{\text{in}}) \tag{9}$$

where $\mathbf{X}_{\text{CAM}}$ is the recalibrated output, $\mathbf{X}_{\text{GM}}$ is the concatenated output of the four VSSS groups from Equation 6, $\mathbf{X}_{\text{in}}$ is the input tensor, and $\text{Affinity}(\mathbf{X}_{\text{in}})$ are the channel-wise attention scores obtained from the channel affinity calculation operation in Equation 8.

## 3.4 DISTILLED LOSS FUNCTION

As mentioned earlier in the motivation in Section 3, the Mamba training is unstable when scaled to large models (Patro & Agneeswaran, 2024). To mitigate this issue, we propose to utilize a distillation objective alongside the standard cross-entropy objective. Knowledge distillation involves training a student model to learn from a teacher model's behavior by minimizing a combination of the classification loss and distillation loss. The distillation loss is computed using the cross-entropy objective between the logits of the teacher and student models. Given the logits ($Z_s$) from the student model, logits ($Z_t$) from a teacher model (RegNetY-16G (Radosavovic et al., 2020) in our case), the ground truth label $y$, and the hard decision of the teacher $y_t = \mathrm{argmax}_c Z_t(c)$, the joint loss function is defined as shown in Equation 10.

$$\mathcal{L}_{\mathrm{total}} = \alpha \mathcal{L}_{\mathrm{CE}}(Z_s, y) + (1 - \alpha)\mathcal{L}_{\mathrm{CE}}(Z_s, y_t). \tag{10}$$

where $\mathcal{L}_{\mathrm{CE}}$ is the cross-entropy objective and $\alpha$ is the weighting parameter. We experimentally show in Section 4 that training with this distillation objective stabilizes training, leading to consistent performance gains on larger model variants.

## 4 EXPERIMENTS

### 4.1 IMAGE CLASSIFICATION

**Settings:** The image classification experiments are based on ImageNet-1K (Deng et al., 2009), which comprising of over $1.28$ million training images and 50K validation images, spanning $1,000$ categories. Following Liu et al. (2022a), we train our models for using the AdamW (Loshchilov & Hutter, 2017) optimizer and a cosine decay learning rate scheduler for 300 epochs, including a 20 epoch warm-up. The total batch size is set to $1024$, with models trained on 8x A100 GPUs, each with 80GB of CUDA memory. Optimizer betas are set to $(0.9, 0.999)$; momentum is set to 0.9, and an initial learning rate of $1 \times 10^{-3}$ is used with a weight decay of 0.05. Label smoothing of 0.1 is used alongside the distillation objective (see Section 3.4).

**Results:** Table 1 presents a comparison of our proposed GroupMamba models (T, S, B) with various state-of-the-art methods. The GroupMamba models exhibit a notable balance of accuracy and computational efficiency. GroupMamba-T achieves a top-1 accuracy of $83.3\%$ with 23 million parameters and $4.5$ GFLOPs, outperforming ConvNeXt-T (Liu et al., 2022b) and Swin-T (Liu et al., 2021) by $1.2\%$ and $2.0\%$, respectively, with fewer parameters. Additionally, GroupMamba-T surpasses the recently introduced SSM models, outperforming VMamba-T (Liu et al., 2024b) and LocalVMamba-T (Huang et al., 2024) by $0.8\%$ and $0.6\%$, respectively, while using $26\%$ fewer parameters than VMamba-T. GroupMamba-S, with 34 million parameters and 7.0 GFLOPs, achieves an accuracy of $83.9\%$, surpassing VMamba-S (Liu et al., 2024b), Swin-S (Liu et al., 2021), and EfficientVMamba-B (Pei et al., 2024). The performance is better than LocalVMamba-S (Huang et al., 2024) by $0.2\%$ with $32\%$ fewer parameters. Furthermore, GroupMamba-B achieves an accuracy of $84.5\%$ with only 57 million parameters and 14 GFLOPs, exceeding VMamba-B (Liu et al., 2024b) by $0.6\%$ while using $36\%$ fewer parameters.

### 4.2 OBJECT DETECTION AND INSTANCE SEGMENTATION

**Settings:** We evaluate the performance of GroupMamba-T for object detection on the MS-COCO 2017 dataset (Lin et al., 2014). Our method is based on the Mask-RCNN (He et al., 2017) detector with the hyperparameters as used for Swin (Liu et al., 2021). We use the AdamW (Loshchilov & Hutter, 2017) optimizer and train Mask-RCNN with GroupMamba-T backbone for 12 epochs. The backbone is initialized and fine-tuned from the ImageNet-1K (Deng et al., 2009). We use an initial learning rate of $1 \times 10^{-4}$ and decay by a factor of 10 at epochs 9 and 11.

**Results:** Table 2 shows the results of GroupMamba-T, comparing it against various state-of-the-art models for object detection and instance segmentation using the Mask R-CNN framework on the MS-COCO dataset. Our model achieves box AP ($AP^b$) of 47.6 and mask AP ($AP^m$) of 42.9. It surpasses ResNet-50 (He et al., 2016), Swin-T (Liu et al., 2022a), ConvNeXt-T (Liu et al., 2022b). In addition, GroupMamba-T has competitive performance compared to VMamba-T (Liu et al., 2024b)

Table 1: **Performance comparison of GroupMamba models with state-of-the-art convolution-based, attention-based, and SSM-based models on ImageNet-1K (Deng et al., 2009)**. Our models demonstrate superior performance and achieve a better trade-off between accuracy and parameters.

| Method | Token mixing | Image size | #Param. | FLOPs | Top-1 acc. |
|---|---|---|---|---|---|
| RegNetY-8G (Radosavovic et al., 2020) | Conv | $224^2$ | 39M | 8.0G | 81.7 |
| RegNetY-16G (Radosavovic et al., 2020) | Conv | $224^2$ | 84M | 16.0G | 82.9 |
| EffNet-B4 (Tan & Le, 2019) | Conv | $380^2$ | 19M | 4.2G | 82.9 |
| EffNet-B5 (Tan & Le, 2019) | Conv | $456^2$ | 30M | 9.9G | 83.6 |
| EffNet-B6 (Tan & Le, 2019) | Conv | $528^2$ | 43M | 19.0G | 84.0 |
| DeiT-S (Touvron et al., 2021) | Attention | $224^2$ | 22M | 4.6G | 79.8 |
| DeiT-B (Touvron et al., 2021) | Attention | $224^2$ | 86M | 17.5G | 81.8 |
| DeiT-B (Touvron et al., 2021) | Attention | $384^2$ | 86M | 55.4G | 83.1 |
| ConvNeXt-T (Liu et al., 2022b) | Conv | $224^2$ | 29M | 4.5G | 82.1 |
| ConvNeXt-S (Liu et al., 2022b) | Conv | $224^2$ | 50M | 8.7G | 83.1 |
| ConvNeXt-B (Liu et al., 2022b) | Conv | $224^2$ | 89M | 15.4G | 83.8 |
| Swin-T (Liu et al., 2021) | Attention | $224^2$ | 28M | 4.6G | 81.3 |
| Swin-S (Liu et al., 2021) | Attention | $224^2$ | 50M | 8.7G | 83.0 |
| Swin-B (Liu et al., 2021) | Attention | $224^2$ | 88M | 15.4G | 83.5 |
| ViM-S (Zhu et al., 2024) | SSM | $224^2$ | 26M | - | 80.5 |
| VMamba-T (Liu et al., 2024b) | SSM | $224^2$ | 31M | 4.9G | 82.5 |
| VMamba-S (Liu et al., 2024b) | SSM | $224^2$ | 50M | 8.7G | 83.6 |
| VMamba-B (Liu et al., 2024b) | SSM | $224^2$ | 89M | 15.4G | 83.9 |
| LocalVMamba-T (Huang et al., 2024) | SSM | $224^2$ | 26M | 5.7G | 82.7 |
| LocalVMamba-S (Huang et al., 2024) | SSM | $224^2$ | 50M | 11.4G | 83.7 |
| EfficientVMamba-B (Pei et al., 2024) | SSM | $224^2$ | 33M | 4.0G | 81.8 |
| GroupMamba-T | SSM | $224^2$ | 23M | 4.5G | 83.3 |
| GroupMamba-S | SSM | $224^2$ | 34M | 7.0G | 83.9 |
| GroupMamba-B | SSM | $224^2$ | 57M | 14G | 84.5 |

and LocalVMamba-T (Huang et al., 2024), with less 20% parameters compared to VMamba-T. Figure 3 (first row) displays qualitative examples of object detection and instance segmentation. GroupMamba-T accurately detects and segments the targets in various scenes.

## 4.3 SEMANTIC SEGMENTATION

**Settings:** We also evaluate the performance of GroupMamba-T for semantic segmentation on the ADE20K (Zhou et al., 2017) dataset. The framework is based on the UperNet (Xiao et al., 2018) architecture, and we follow the same hyperparameters as used for the Swin (Liu et al., 2021) backbone. More specifically, we use the AdamW (Loshchilov & Hutter, 2017) optimizer for a total of $160k$ iterations with an initial learning rate of $6 \times 10^{-5}$. The default input resolution used in our experiments is $512 \times 512$.

**Results:** The GroupMamba-T model demonstrates favorable performance in semantic segmentation compared to various state-of-the-art methods, as presented in Table 3. GroupMamba-T achieves a mIoU of $48.6$ in single-scale and $49.2$ in multi-scale evaluation, with $49M$ parameters and $955G$ FLOPs. This outperforms ResNet-50 (He et al., 2016), Swin-T (Liu et al., 2021), and ConvNeXt-T (Liu et al., 2022b). Additionally, GroupMamba-T exceeds the performance of the recent SSM methods, including ViM-S (Zhu et al., 2024), VMamba-T (Liu et al., 2024b), and LocalVMamba (Huang et al., 2024) with fewer number of parameters. Figure 3 (second row) shows qualitative examples of GroupMamba-T. These examples demonstrate our model's ability to accurately segment various classes for indoor and outdoor scenes.

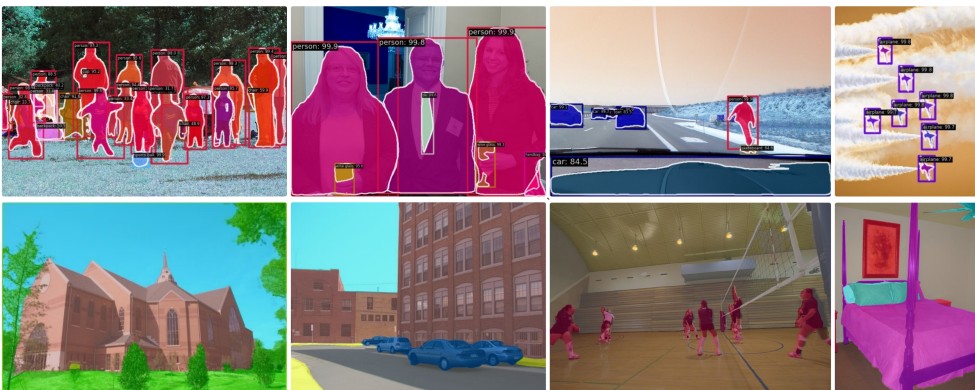

Figure 3: Qualitative results of GroupMamba-T for object detection and instance segmentation (first row) on the MS-COCO val. set and semantic segmentation (second row) on ADE20k val. set.

Table 2: **Performance comparison for object detection and instance segmentation on MS-COCO (Lin et al., 2014) using Mask R-CNN (He et al., 2017):** $AP^b$ and $AP^m$ signify box AP and mask AP, respectively. FLOPs, are computed for an input dimension of $1280 \times 800$.

| Mask R-CNN 1× schedule | | | | | | | | |
|---|---|---|---|---|---|---|---|---|
| Backbone | $AP^b$ | $AP^b_{50}$ | $AP^b_{75}$ | $AP^m$ | $AP^m_{50}$ | $AP^m_{75}$ | #param. | FLOPs |
| ResNet-50 (He et al., 2016) | 38.2 | 58.8 | 41.4 | 34.7 | 55.7 | 37.2 | 44M | 260G |
| Swin-T (Liu et al., 2021) | 42.7 | 65.2 | 46.8 | 39.3 | 62.2 | 42.2 | 48M | 267G |
| ConvNeXt-T (Liu et al., 2022b) | 44.2 | 66.6 | 48.3 | 40.1 | 63.3 | 42.8 | 48M | 262G |
| PVTv2-B2 (Wang et al., 2022) | 45.3 | 67.1 | 49.6 | 41.2 | 64.2 | 44.4 | 45M | 309G |
| VMamba-T (Liu et al., 2024b) | 47.4 | 69.5 | 52.0 | 42.7 | 66.3 | 46.0 | 50M | 270G |
| LocalVMamba-T (Huang et al., 2024) | 46.7 | 68.7 | 50.8 | 42.2 | 65.7 | 45.5 | 45M | 291G |
| GroupMamba-T | 47.6 | 69.8 | 52.1 | 42.9 | 66.5 | 46.3 | 40M | 279G |

Table 3: **Performance comparison for semantic segmentation on ADE20K (Zhou et al., 2017) using UperNet (Xiao et al., 2018).** The terms 'SS' and 'MS' refer to evaluation at single-scale and multi-scale levels, respectively. FLOPs are computed for an input dimension of $512 \times 2048$.

| method | crop size | mIoU (SS) | mIoU (MS) | #param. | FLOPs |
|---|---|---|---|---|---|
| ResNet-50 (He et al., 2016) | $512^2$ | 42.1 | 42.8 | 67M | 953G |
| Swin-T (Liu et al., 2021) | $512^2$ | 44.4 | 45.8 | 60M | 945G |
| ConvNeXt-T (Liu et al., 2022b) | $512^2$ | 46.0 | 46.7 | 60M | 939G |
| ViM-S (Zhu et al., 2024) | $512^2$ | 44.9 | - | 46M | - |
| VMamba-T (Liu et al., 2024b) | $512^2$ | 48.3 | 48.6 | 62M | 948G |
| EfficientVMamba-B (Pei et al., 2024) | $512^2$ | 46.5 | 47.3 | 65M | 930G |
| LocalVMamba-T (Huang et al., 2024) | $512^2$ | 47.9 | 49.1 | 57M | 970G |
| GroupMamba-T | $512^2$ | 48.6 | 49.2 | 49M | 955G |

## 4.4 ABLATION STUDY

Figure 4 showcases the impact of each proposed contribution in terms of top-1 accuracy, number of parameters, and throughput, compared to other SSM-based methods. GroupMamba-T with 4-D scanning, comprising 22M parameters, achieves a top-1 accuracy of 82.30% and a throughput of 803. By applying a unidirectional 1D scan across $N/4$ channels in four directions—left-to-right, right-to-left, top-to-bottom, and bottom-to-top instead of the full 4-D scanning across all $N$ channels, the throughput significantly increased from 803 to 1125, with only a negligible accuracy reduction of 0.1%, while keeping the same number of parameters.

The integration of the CAM module further elevates the top-1 accuracy from 82.20% to 82.50%, with a minor reduction in throughput (from 1125 to 1069). Finally, incorporating the proposed distillation-based loss pushes the top-1 accuracy to 83.30%, while preserving the throughput at 1069.

In comparison to Vim-S (Zhu et al., 2024), GroupMamba has fewer parameters and outperforms it by 2.8% in top-1 accuracy, with 1.5× higher throughput. When compared to LocalVMamba-T (Huang et al., 2024), GroupMamba achieves a 0.5% gain in top-1 accuracy while being 3× faster and having fewer parameters. Compared to VMamba-T (Liu et al., 2024b), our model demonstrates slightly faster throughput, a 0.6% increase in top-1 accuracy, and a 26% improvement in parameter efficiency.

To demonstrate the training stability of GroupMamba-Base variant compared to the baseline VMamba-Base, we evaluate the loss progression and variance throughout the training process. For the baseline variant, the initial loss at epoch 0 was 6.9325 and decreased to 2.2021 (2.4731) by epoch 300, with a variance of 0.67142. In contrast, GroupMamba-Base exhibited a starting loss of 6.9272, which dropped to 1.2651 (1.4827) by epoch 300, accompanied by a lower variance of 0.46916. This indicates enhanced training stability for GroupMamba-Base, showcasing better convergence compared to the baseline VMamba-Base.

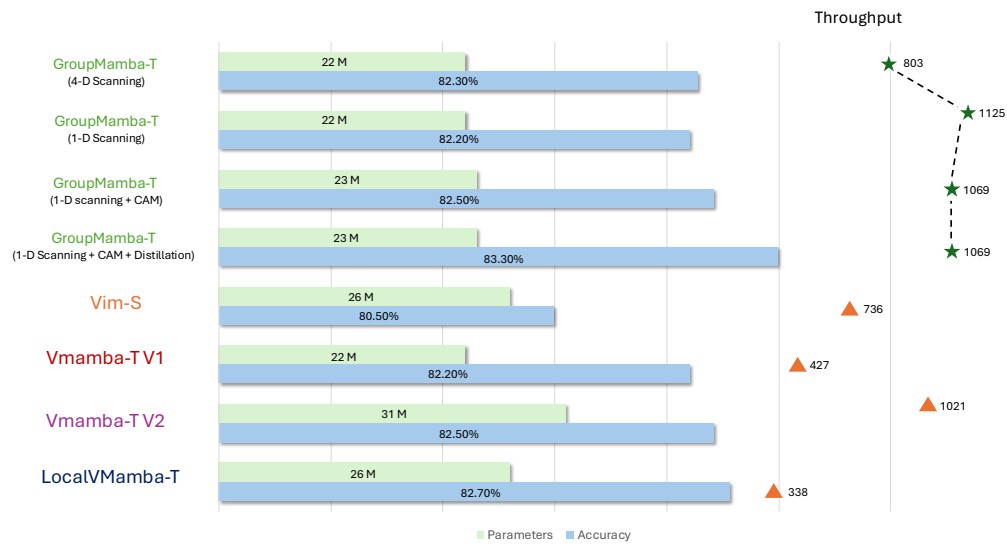

Figure 4: Comparison of GroupMamba variants and SSM-based methods in classification accuracy and computational efficiency. The throughput (number of predicted samples per second) is measured using a single Nvidia A100 GPU with a batch size of 128 for all methods.

## 5 CONCLUSION AND FUTURE WORK

In this paper, we tackle the computational inefficiencies and stability challenges associated with visual SSMs for computer vision tasks by introducing a novel layer called the *Modulated Group Mamba*. We also propose a multi-directional scanning method that improves parameter efficiency by scanning in four spatial directions and leveraging the *Channel Affinity Modulation* (CAM) operator to enhance feature aggregation across channels. To stabilize training, especially for larger models, we employ a distillation-based training objective. Our experimental results demonstrate that the proposed GroupMamba models outperform recent SSMs while requiring fewer parameters.

Our research has focused on image classification, object detection, instance segmentation, and semantic segmentation. To further validate and extend the generalization ability of our method, we aim to explore additional downstream tasks, such as video recognition and time-series data applications. Evaluating the Modulated Group Mamba layer in these contexts will help to uncover its potential benefits and limitations, providing deeper insights and guiding further improvements.

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
