# A    SUPPLEMENTARY MATERIAL

This section contains supplemental material, offering further results and analysis to complement the main paper. We provide additional details on the following topics:

- Architectural Details (Section A.1)

- Ablations (Section A.2)

- Qualitative Results (Section A.3)

- Discussion (Section A.4)

## A.1    ARCHITECTURAL DETAILS

We develop three variants of our GroupMamba backbones, each tailored to different performance and efficiency requirements: GroupMamba-T (Tiny), GroupMamba-S (Small), and GroupMamba-B (Base). These variants differ in their channel dimensions and the number of layers per stage, as detailed in Table 4.

Table 4: GroupMamba Architectures. Description of the configurations of the model variants for the embedding size, the number of layers, and the model's GFLOPs and Parameters. Between two consecutive stages, we incorporate a downsampling layer to increase the number of channels and reduce the resolution by two.

| Stage | Output Resolution | Type | Config | GroupMamba | | |
|---|---|---|---|---|---|---|
| | | | | T | S | B |
| Stem | $\frac{H}{2} \times \frac{W}{2}$ | Patch Embedding | Patch Size | k=3x3, s=2 | | |
| | | | Embed. Dim. | 32 | 32 | 48 |
| | $\frac{H}{4} \times \frac{W}{4}$ | Patch Embedding | Patch Size | k=3x3, s=2 | | |
| | | | Embed. Dim. | 64 | 64 | 96 |
| 1 | $\frac{H}{4} \times \frac{W}{4}$ | Modulated Group Mamba | Layers | 3 | 3 | 3 |
| | $\frac{H}{8} \times \frac{W}{8}$ | Down-Sampling | Patch Size | k=3x3, s=2 | | |
| | | | Embed. Dim. | 128 | 128 | 192 |
| 2 | $\frac{H}{8} \times \frac{W}{8}$ | Modulated Group Mamba | Layers | 4 | 4 | 6 |
| | $\frac{H}{16} \times \frac{W}{16}$ | Down-Sampling | Patch Size | k=3x3, s=2 | | |
| | | | Embed. Dim. | 320 | 320 | 384 |
| 3 | $\frac{H}{16} \times \frac{W}{16}$ | Modulated Group Mamba | Layers | 6 | 12 | 18 |
| | $\frac{H}{32} \times \frac{W}{32}$ | Down-Sampling | Patch Size | k=3x3, s=2 | | |
| | | | Embed. Dim. | 448 | 512 | 512 |
| 4 | $\frac{H}{32} \times \frac{W}{32}$ | Modulated Group Mamba | Layers | 3 | 3 | 3 |
| Parameters | | | | 23M | 34M | 57M |
| FLOPs | | | | 4.5G | 7.0G | 14.0G |

## A.2    ABLATIONS

In Table 5, we provide additional ablation results regarding the distillation training objective. For the GroupMamba-T and GroupMamba-S variants, the distilled loss improves performance by an absolute gain of 0.8% and 0.9%, respectively. For the largest variant, GroupMamba-B, the distilled loss improves performance by 1.3%. This demonstrates that larger Mamba-based models with MLP tend to saturate and struggle to converge effectively without distillation. Incorporating distillation for the large model boosts its performance from 83.2% to 84.5%.

Table 5: Ablation study on GroupMamba variants with and without the Distilled Loss.

| Method | #Param. | FLOPs | Top-1 acc. |
|---|---|---|---|
| GroupMamba-T w/o Distilled Loss | 23M | 4.6G | 82.5 |
| GroupMamba-T with Distilled Loss | 23M | 4.6G | 83.3 (+0.8) |
| GroupMamba-S w/o Distilled Loss | 34M | 7.0G | 83.0 |
| GroupMamba-S with Distilled Loss | 34M | 7.0G | 83.9 (+0.9) |
| GroupMamba-B w/o Distilled Loss | 57M | 14G | 83.2 |
| GroupMamba-B with Distilled Loss | 57M | 14G | 84.5 (+1.3) |

## A.3 QUALITATIVE RESULTS

In Figure 5, we present the qualitative results of GroupMamba-T on samples from the COCO validation set Lin et al. (2014), demonstrating its performance in instance segmentation and object detection. Our model accurately localizes objects and correctly segments them across diverse scenes and varying scales. In Figure 6, we show additional qualitative results of GroupMamba-T on samples from the ADE20K Zhou et al. (2017) validation set for semantic segmentation. The first row shows the ground truth masks, while the second row displays the predicted masks. It is notable that our model delineates the masks accurately, highlighting the effectiveness for semantic segmentation. The quantitative and qualitative results of GroupMamba demonstrate the robust generalization capability of our GroupMamba backbones across diverse downstream tasks, including semantic segmentation, object detection, and instance segmentation.

## A.4 DISCUSSION

Our main contributions include introducing the Modulated Group Mamba layer, which enhances computational efficiency and interaction in state-space models through a multi-direction scanning method. We also introduce the Channel Affinity Modulation (CAM) operator to improve feature aggregation across channels, addressing limitations in grouping operations. Additionally, we employ a distillation-based training objective to stabilize the training of models with a large number of parameters. These contributions enable us to achieve competitive performance with recent state-space models in image classification, object detection, instance segmentation, and semantic segmentation with fewer number of parameters.

This can further facilitate the development of vision foundation models based on Mamba that can be scaled to a large number of parameters efficiently and stably. The Modulated Group Mamba layer and CAM operator enhance computational efficiency and feature interaction, allowing models to manage more extensive and complex datasets without excessive resource demands. The distillation-based training objective ensures stability during training, which is crucial for maintaining performance as model sizes increase. Together, these advancements enable the creation of scalable, reliable vision models that can be deployed effectively in various real-world applications.

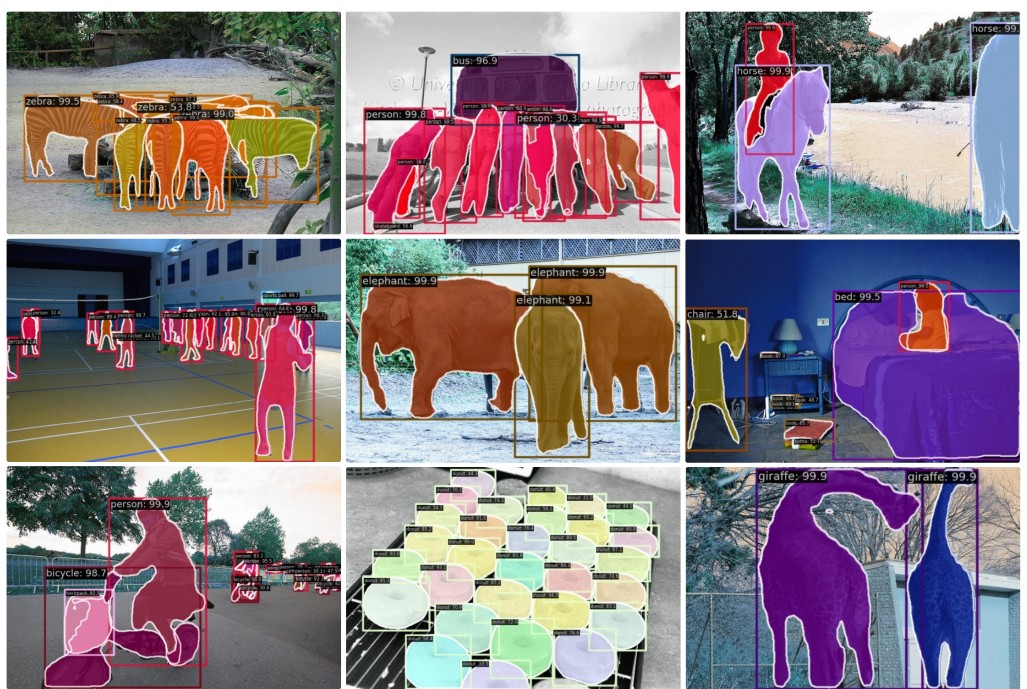

Figure 5: Qualitative results of GroupMamba-T for object detection and instance segmentation on the COCO validation set.

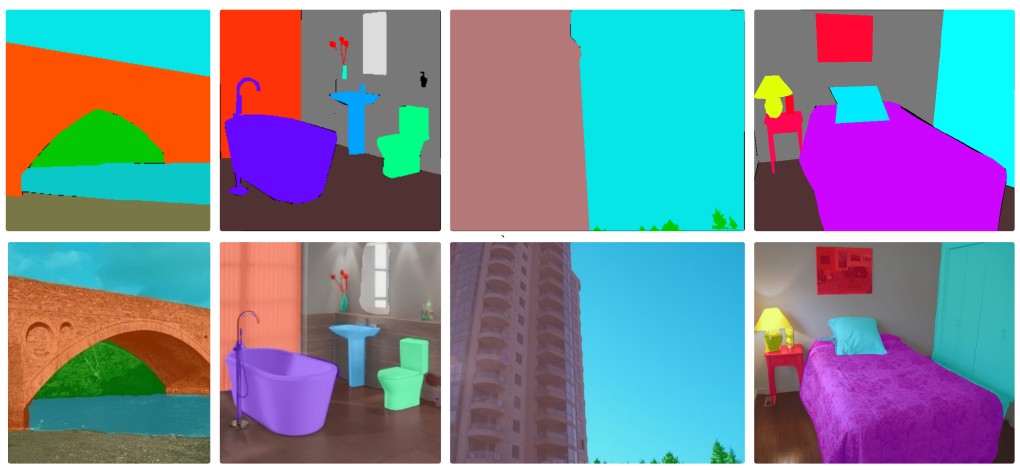

Figure 6: Qualitative results of GroupMamba-T for semantic segmentation on ADE20K validation set. The first row shows the ground truth for the masks, while the second and second show the corresponding predictions of our model.