# OpenReview forum: "GroupMamba: Parameter-Efficient and Accurate Group Visual State Space Model"
_ICLR.cc/2025/Conference — ICLR 2025 Conference Withdrawn Submission_

### Official Review · Reviewer_vP76 · 2024-10-27

**Soundness:** 2
**Presentation:** 2
**Contribution:** 1
**Rating:** 3
**Confidence:** 5

**Summary:**

This paper introduces a group scanning strategy inspired by group convolution. Channel attention is used for channel modulation. Additionally, this paper adopts distillation techniques to improve performance.

**Strengths:**

1. The proposed channel modulation module somewhat improves performance, and this paper shows that distillation techniques could bring significant improvements to vision mamba models.

**Weaknesses:**

1. Writing problems:
    1) The introduction section could be improved. I cannot get an overview of how this paper addresses the proposed problem. There is almost no description of the proposed methods.
    2) There are factual errors in the given equation. For example, in Lines 209-210, the dimensionality of **C** is incorrect. It should be 1 times N instead of N times 1.

2. The motivation is outdated. The authors claim that vision state space models lack stability when scaling to larger models. This motivation may not be accurate. Although pioneer vision mamba works such as ViM and VMambav1 suffer from unstable training when scaling the model to base size, subsequent optimizations in hyperparameters and implementations have addressed this issue. Recent vision mamba works, such as VMambav3, have achieved stable training for base-size models.

3. The proposed "GROUPED MAMBA OPERATOR" has significant overlap with previous work, UltraLight VM-UNet[1], which was released on Arxiv on 24.03.08. The implementation is almost identical.

4. The comparison in Table 1 is unfair. As GroupMamba uses distillation to improve performance, it should also compare with methods that use the same distillation technique, or it should report the performance without distillation for a fair comparison. Under fair comparison, the proposed GroupMamba lags behind current vision mamba works, such as GrootVL[2], by a large margin.

5. The experiment is not sufficient. A larger model and a longer training schedule on the COCO detection dataset are required.

Reference:

[1]. Wu, Renkai, et al. "Ultralight vm-unet: Parallel vision mamba significantly reduces parameters for skin lesion segmentation." arXiv preprint arXiv:2403.20035 (2024).

[2]. Xiao, Yicheng, et al. "GrootVL: Tree Topology is All You Need in State Space Model." arXiv preprint arXiv:2406.02395 (2024).

**Questions:**

Could the authors provide performance data on the COCO dataset for both small and base sizes, with both 1x and 3x training schedules?

---

### Official Review · Reviewer_iBCX · 2024-10-31

**Soundness:** 3
**Presentation:** 2
**Contribution:** 2
**Rating:** 3
**Confidence:** 5

**Summary:**

The paper proposes a grouped parameter-efficient State Space Model (SSM) for computer vision tasks, named GroupMamba. The authors introduce a Modulated Group Mamba layer that divides input channels into different groups and utilize a Channel Affinity Modulation (CAM) operator to enhance the cross-channel communication. The authors also propose a distilled loss to stabilize large SSMs training. Experimental results show that the proposed method achieves superior performance compared to existing methods.

**Strengths:**

1. In GroupMamba, the design of grouping different scanning directions significantly reduces the parameter count. Additionally, by incorporating a Channel Affinity Module and a distillation loss, it achieves high performance while maintaining efficiency. These enhancements make GroupMamba well-suited for deployment in resource-limited environments.

**Weaknesses:**

1. The novelty is limited. The Modulated Group Mamba layer appears to be a channel-reduced version of VMamba. From an implementation perspective, VMamba uses four scanning directions, with SSM module in each direction having a channel of
C. In contrast, GroupMamba reduces the channel to C/4 and introduces a Channel Affinity Module (CAM) to integrate affinity between channels. To further establish the innovation of this approach, please clarify the fundamental differences between this method and a channel-reduced VMamba, or provide an analysis of the trade-offs and advantages of this design choice, so the innovation and practical value of GroupMamba can be better understood.

2. The novelty of CAM module is limited. The CAM module proposed in the article is almost identical to the Squeeze-and-Excitation (SE) block in the paper [1]. Both CAM and SE blocks adjust channel importance by recalibrating features at the channel level. SE blocks do this by applying a scaling factor to each channel after using a global pooling operation to extract channel-wise information. CAM uses a similar approach to refine the importance of each channel through affinity calculations. Please clarify the differences between the CAM and SE block within the context of State Space Models (SSM) to demonstrate the novelty of the approach. Eq. (3) in paper [1] and Eq. (8) in this paper have almost identical formulations, indicating that the authors are aware of this paper. Therefore, including the necessary citations is essential.

3. Performance improvement and distillation loss: It appears that the performance gains are largely attributed to the use of distillation loss (evident in the classification improvement from 82.5% to 83.3% on GroupMamba-T), a widely recognized technique for boosting lightweight classification model performance. Excluding the distillation loss, GroupMamba achieves results comparable to the latest Mamba models, such as VMamba and LocalMamba. However, it remains a parameter-efficient Mamba architecture. Please provide the specific performance of GroupMamba on detection and segmentation tasks without using distillation loss.

4. Throughput. While this approach reduces model parameters, the model's efficiency is significantly slower than the baseline, with VMamba v2 reporting a throughput of 1340. Please explain more about the gap between throughput.

Reference:
[1]	Hu, Jie, Li Shen, and Gang Sun. "Squeeze-and-excitation networks." Proceedings of the IEEE conference on computer vision and pattern recognition. 2018.

**Questions:**

1. The experiments on downstream tasks, such as detection and segmentation, are currently incomplete. V2M has only been tested on the tiny variant, and additional results for the small and base variants should be provided.

2. Figure 3 only shows the visual results of the proposed method on detection or segmentation tasks, lacking qualitative comparison with other methods, such as VMamba or LocalMamba.

3. Writing Quality: The overall clarity and writing quality of the paper would benefit from further refinement. 1) Please use clearer mathematical notation to formulate the equations in Sec. 3.3, especially Eq. (3), (6) and (9) so that they align with the form of Eq. (2) in Sec 3.1.

---

### Official Review · Reviewer_PNv8 · 2024-11-03

**Soundness:** 2
**Presentation:** 3
**Contribution:** 1
**Rating:** 3
**Confidence:** 5

**Summary:**

This paper presents a vision of mamba architecture for general visual recognition. Specifically, certain engineering modifications have been added to the recent VMamba architecture, leading to quantitative improvements to the commonly used visual recognition benchmarks.

**Strengths:**

1. The paper is clearly written and easy to understand.

**Weaknesses:**

1. This paper's second claimed technical contribution, the Channel Affinity Modulation (CAM), is very similar, if not identical, to the Squeeze-and-Excitation block [a]. However, the paper neither mentions nor cites [a]. It is unclear if this omission was intentional, but I highly believe that authors are aware of the existence of [a] because 1) [a] is well-known by the community and has gained over 35k citations, 2) the formulation in Section 3.3.3 is very similar to the formulations in [a] (especially the usage of $\sigma$ and $\delta$ in Equation (8)). Therefore, the proposed channel affinity modulation should not be recognized as a technical contribution. On the other hand, for the sake of academic integrity, I strongly recommend that the authors cite [a].

2. The novelty of the Grouped Mamba Operator is also limited. Although technically sound, it should be viewed more as an engineering refinement rather than an original contribution, as it is only about modifying the baseline VMamba's SSM module by reducing the channel numbers and decoupling the SSM parameters.

3. All experimental advances of the proposed method are overclaimed because of the usage of knowledge distillation because all competing models in the experiments are trained without knowledge distillation. For a fair evaluation, the authors should compare their method against models distilled using the same teacher network, RegNetY-16G.

4. I agree that improving Vision Mamba's scalability is an important challenge in developing the next generation of vision SSMs. While the paper argues that its distillation approach enhances the scalability of the Vision Mamba's poor scaling ability, it lacks clear motivation and supporting evidence, such as theoretical analysis or quantitative ablation studies. The claim that "accuracy improves with distillation" is insufficient, as knowledge distillation generally boosts accuracy. Therefore, Providing more substantial evidence is necessary to support this claim.

[a] Hu, Jie, Li Shen, and Gang Sun. "Squeeze-and-excitation networks." Proceedings of the IEEE conference on computer vision and pattern recognition. 2018.

**Questions:**

Why do the fonts in some parts of the manuscript differ from the official template, especially the formulation fonts?

---

### Note · Authors · 2024-11-12

I have read and agree with the venue's withdrawal policy on behalf of myself and my co-authors.